# Five New Terpenes with Cytotoxic Activity from *Pestalotiopsis* sp.

**DOI:** 10.3390/molecules26237229

**Published:** 2021-11-29

**Authors:** Dan Zhao, Meigeng Hu, Guoxu Ma, Xudong Xu

**Affiliations:** Key Laboratory of Bioactive Substances and Resource Utilization of Chinese Herbal Medicine, Ministry of Education, Key Laboratory of Innovative Drug Discovery of Traditional Chinese Medicine (Natural Medicine) and Translational Medicine, Institute of Medicinal Plant Development, Peking Union Medical College and Chinese Academy of Medical Sciences, Beijing 100193, China; zhaodanonline@foxmail.com (D.Z.); mghu@implad.ac.cn (M.H.)

**Keywords:** *Pestalotiopsis*, *Ligusticum chuanxiong*, terpenes, cytotoxic activities, HuH-7 cell lines, lung cancer

## Abstract

Five new compounds called Pestalotis A–E (**1**–**5**), comprising three monoterpene-lactone compounds (**1**–**3**), one tetrahydrobenzofuran derivative (**4**), and one sesquiterpene (**5**), were isolated from the EtOAc extract of *Pestalotiopsis* sp. The structures of the new compounds were elucidated by analysis of their NMR, HRMS, and ECD spectra, and the absolute configurations were established through the comparison of experimental and calculated ECD spectra. All compounds were tested for antitumor activity against SW-480, LoVo, HuH-7, and MCF-7. The results showed that compounds **2** and **4** exhibited potent antitumor activity against SW-480, LoVo, and HuH-7 cell lines. Furthermore, compound **4** was assessed against HuH-7, and the results indicated that the rate of apoptosis was dose-dependent.

## 1. Introduction

Secondary metabolites are natural products that have been selected by evolution over millions of years, displaying a unique chemical diversity and corresponding diversity of biological activities [1,2,3]. There is increasing evidence that many of the important drugs originally thought to be produced by plants are probably products of an interaction with endophytic microbes residing in the tissues between living plant cells [4,5,6]. The microbial universe clearly presents a vast untapped resource for drug discovery [7,8]. *Pestalotiopsis* sp. belongs to the genus of asexual endophytic fungi of seminomycetes, and has received much attention for its production of structurally diverse and complex molecules [9]. In previous research, investigations of the fungi of *Pestalotiopsis* genus have made significant progress in new structure exploration, which has resulted in the discovery of various types of compounds including terpenoids, alkaloids, polyketones, and anthraquinones [10,11], and these have exhibited a broad spectrum of, among others, antitumor, anti-virus, anti-bacterial, and immunosuppressive activities [12,13]. Recent chemical research on the species resulted in the production of many terpenoid molecules with new structures and obvious biological activities, such as structurally complex epoxyquinols [14], polyketide-terpene hybrid metabolites with highly functionalized groups [15], and unusual new bicyclic and tricyclic sesquiterpenoids that bear potent immunosuppressive and cytotoxic activity [12]. Motivated by its fascinating secondary metabolites, we launched a chemical investigation of both new and bioactive compounds from *Pestalotiopsis* sp., which is an endophytic fungus derived from the surface-sterilized root of *Ligusticum chuanxiong* Hort. The results showed significant antiproliferative activities against several cancer cell lines. Therefore, a chemical investigation was conducted, leading to the discovery of three new monoterpenes Pestalotis A–C, one new tetrahydrobenzofuran Pestalotis D, and one sesquiterpene Pestalotis E (Figure 1), among which compounds **1** and **4** showed potent effects. Herein, we report the isolation, structure, and bioactivities of these compounds.

## 2. Results

Compound **1** was obtained as a white amorphous powder. It was assigned the molecular formula C_15_H_20_O_5_, based on an HRESIMS analysis (ion peak at m/z 303.1210 [M + Na]^+^, calcd for C_15_H_20_O_5_Na 303.1203), indicating 6 degrees of unsaturation. The IR absorption bands at 3401, 1646, and 1633 cm^−1^ suggested the presence of hydroxyl and carbonyl. The ^1^H NMR data (Table 1) showed signals for four sets of methyl protons at δ_H_ 2.14 (3H, s, H-4′), 1.93 (3H, s, H-5′), 1.74 (3H, s, H-10), and δ_H_ 1.03 (3H, s, H-11), two methylene protons at δ_H_ 2.63 (1H, m, H-4a), 2.45 (1H, m, H-4b), 1.87 (1H, m, H-5a), and 1.54 (1H, m, H-5b), and two oxygenated methines at δ_H_ 4.51 (1H, d, J = 9.6, H-7) and 4.97 (1H, d, J = 9.6, H-8), which could be easily discerned. Moreover, 15 carbon resonances were revealed in the ^13^C NMR spectra, and assigned to four methyl groups (δ_C_ 20.1, 27.0, 8.3, and 25.2), and six quaternary carbons comprising two ester carbonyls (δ_C_ 174.0, 165.0), three olefinic carbons (δ_C_ 120.1, 160.0, and 158.5), and one oxygenated quaternary carbon (δ_C_ 71.0). The above structural features pointed to a monoterpene scaffold for **1** [16].

Further analyses of HMBC spectra (Figure 2) led to the elucidation of the planar structure of **1**. The long-range heteronuclear couplings from Me-4′ and Me-5′ to the olefinic double-bond carbons C-2′ and C-3′ secured the connectivity from C-2′ to C-3′ bearing these dimethyl groups. Strong HMBC correlations from H-2′ and H-3′ to the carbonyl carbon C-1′ at δ_C_ 165.0 extended an isobutene attached to C-1′. Additionally, the key HMBC correlation from H-5 to C-1′ and from H-7 to C-1′ revealed that isopentanoic acid was connected to C-6. The methyl group (Me-10) was adjacent to C-3 based on the HMBC correlations from H-10 to C-3 and one carbonyl carbon C-2, and of H-11 to C-5 and C-6, suggesting the connectivity of the methyl carbonyl group (Me-11) attached to C-6.

A NOESY experiment (Figure 3) was conducted to elucidate the relative configuration of **1**. The cross-peaks of Me-11/H-7 demonstrated that H-7 and Me-11 were cofacial and α-oriented. Moreover, the large coupling constants (J = 9.6 Hz) of H-7 with vicinal protons demanded the trans-orientation of the substituents, which was verified by the NOE correlations of H-7/H-8. To further elucidate its absolute configuration, the ECD spectra of **1** and ent-**1** were calculated, and the former was identical to the experimental ECD curve, suggesting that its absolute configuration was 6S,7R,8S. The structure was calculated using the TDDFT methodology at the PBE0/def2-TZVP level in MeOH. Therefore, the structure of compound **1** was identified as shown and given the trivial name Pestalotis A.

Compound **2** was obtained as a white powder with the molecular formula of C_15_H_22_O_5_, in agreement with the positive HRESIMS ion peak at m/z 305.1387 [M + Na]^+^ (calcd for C_15_H_22_O_5_Na 305.1359). An intensive comparison of its ^1^H and ^13^C spectra with those of **1** indicated that compound **2** was closely similar to **1**. The noticeable chemical shift differences as compared to **1** were detected for H-2′ (δ_H_ 5.81 to δ_H_ 2.33) with additional signals H-3′ (δ_H_ 2.08) and for the corresponding carbons being shielded from δ_C_ 115.1 to δ_C_ 42.7 (C-2′) and δ_C_ 158.5 to δ_C_ 25.3 (C-3′), which indicated that the double bonds at C-2′ and C-3′ of compound **2** were reduced. The HMBC correlations from H-3′ to C-2′, C-4′, and C-5′ further demonstrated the above deduction. Its NOESY correlations and ECD spectrum were almost the same as those of **1**. Hence, the absolute configuration of **2** was determined to be 6S,7R,8S and named Pestalotis B.

Compound **3** had a molecular formula of C_15_H_26_O_5_ based on the ^13^C NMR data and HRESIMS ion at m/z 307.1500 [M + Na]^+^ (calcd for C_15_H_26_O_5_Na 307.1516). The NMR data (Table 1) of compound **3** were similar to those of compound **2**. The major differences were that two sp^3^ methines δ_H_ 2.11 (1H, m, H-3) and 1.90 (1H, m, H-9) were present in **3**, as supported by the 2D NMR analyses. Furthermore, the key HMBC correlations from H-3 to C-10, C-2, and C-9 (δ_C_ 36.9), and H-9 to C-8 and C-3 (δ_C_ 40.9) implied that the double bonds of H-2 and H-3 were reduced. The NOESY NMR data of **3** were similar to those of **1**, suggesting that they shared the same relative configuration. Furthermore, by combined ECD calculations, the absolute configuration of **3** was determined to be 6S,7R,8S and named Pestalotis C.

Compound **4** showed a protonated molecule [M + Na]^+^ at m/z 346.1627 (calcd for C_17_H_25_NO_5_Na 346.1625), indicating a molecular formula C_17_H_25_NO_5_. Its ^1^H NMR spectrum (Table 2) displayed signals characteristic of four methyl groups δ_H_ 0.92 (3H, d, J = 2.4 Hz, H-13), 0.91 (3H, d, J = 2.4 Hz, H-14), 2.03 (3H, s, H-17), and 1.13 (3H, s, H-18), seven methylene groups δ_H_ 2.51(1H, m, H-4a), 2.30 (1H, m, H-4b), 1.84 (1H, m, H-5a), 1.59 (1H, m, H-5b), 2.70 (2H, dd, J = 1.8, 7.2 Hz, H-11), 5.28 (2H, d, J = 13.2 Hz, H-15a), and 5.22 (1H, d, J = 13.2 Hz, H-15b), and an oxygenated methine δ_H_ 4.22 (1H, s, H-7). The ^13^C and HSQC NMR spectra (Figure 2) of **4** revealed 16 carbon resonances, comprising four methyls (δ_C_ 22.5, 22.4, 20.4, and 24.9), four methylenes (δ_C_ 17.6, 32.8, 47.1, and 56.9), two methines (δ_C_ 24.1 and one oxygenated at δ_C_ 68.6), four sp^2^ quaternary carbons (δ_C_ 147.6, 126.4, 121.6, and 154.8), an oxygenated quaternary carbon (δ_C_ 70.4), and a conjugated ketone (δ_C_ 190.4). The aforementioned information, along with the reported (6S,7S)-6,7-dihydroxy-3,6-dimethyl-2-isovaleroyl-4,5,6,7-tetrahydrobenzofuran, suggested that **4** was closely related to tetrahydrobenzofuran-type [17]. Then, one nitrogen atom in the molecular formula was identified as an amide (at δ_C_ 170.1) among the carbonyl group, as supported by the HMBC correlation from H-15 to δ_C_ 126.4 (C-3), 147.6 (C-2), and 121.6 (C-9). The presence of an isovaleroyl group was determined by the ^1^H-^1^H COSY correlations of H-11/H-12/Me-13/Me-14, as well as HMBC cross-peaks from 2.70 (2H, dd, J = 1.8, 7.2 Hz, H-11) to C-10 (δ_C_ 190.4), C-12 (δ_C_ 24.1), C-13 (δ_C_ 22.5), C-14 (δ_C_ 22.4), and C-2 (δ_C_ 147.6), from 0.92 (3H, d, J = 2.4 Hz, H-13) and 0.91 (3H, d, J = 2.4 Hz, H-14) to C-12 and C-11 (δ_C_ 47.1), and from 2.70 (2H, dd, J = 1.8, 7.2 Hz, H-11) to C-10 (δ_C_ 190.4), which demonstrated the attachment of the isovaleroyl group to C-2. In addition, COSY data are also supported by the connection of two other methylenes H-4 and H-5, which constituted a part of the six-membered ring (Figure 2). Meanwhile, the location of the methyl group was confirmed as being attached at C-6 by the HMBC correlations from Me-18 to C-5, C-6, and C-7. Thus, the planar structure of **4** was determined. The absolute configuration of compound **4** was further confirmed by the calculated ECD spectrum at the PBE0/def2-TZVP level being consistent with its experimental spectrum, allowing its absolute configuration to be confidently assigned as 6S,7S. As a result, the structure of **4** was established and named Pestalotis D.

Compound **5** gave the molecular formula C_17_H_26_O_4_ by its positive HRESIMS at m/z 317.1725 [M+Na]^+^, indicating 5 degrees of unsaturation. Its ^1^H NMR spectrum (Table 2) displayed signals characteristic of one olefinic proton δ_H_ 7.01 (1H, d, J = 6.0 Hz, H-5), three sets of diastereotopic protons at δ_H_ 2.63 (2H, m, H-8), 1.69 (1H, m, H-2a), and 1.45 (1H, m, H-2b) and δ_H_ 1.54 (1H, m, H-3a), 1.34 (1H, m, H-3b), and signals attributable to methylene at δ_H_ 2.41 (2H, m, H-12), four methine protons at δ_H_ 1.84 (1H, m, H-4), 2.12 (1H, m, H-11), 2.22 (1H, m, H-9), and 2.49 (1H, m, H-10), methyl group protons at δ_H_ 0.99 (3H, d, J = 7.2 Hz, H-13), 1.77 (3H, s, H-14), 1.30 (3H, s, H-15), and 2.01 (3H, s, H-17), as confirmed by multiplicity-edited HSQC data. The ^13^C NMR spectrum indicated the presence of 15 magnetically nonequivalent carbons divided into two carbonyls (δ_C_ 202.6 and 173.1), one oxygenated tertiary carbon (δ_C_ 71.8), an olefinic group (δ_C_ 151.2 and 136.7), four methylenes (δ_C_ 34.7, 23.1, 36.4, and one oxygenated at 69.4), two methines (δ_C_ 40.1 and 33.7), and four methyls (δ_C_ 11.1, 16.2, 27.6, and 21.0). The ^1^H and ^13^C NMR spectroscopic data of **5** confirmed that the compound was a sesquiterpenes alcohol [18]. The positions of the α,β-unsaturated carbonyl group were suggested by the HMBC correlations from Me-14 to C-5 and C-7. Finally, the HMBC correlation from Me-17 to C-16 and C-12, and from H-12 to C-16, suggested the presence of the methyl ester moiety, which was assigned at C-12 to complete the planar structure of **5** (Figure 2). A NOESY experiment was conducted to elucidate the relative configuration of **5** (Figure 3). The cross-peaks of Me-15 (δ_H_ 1.30)/H-10 (δ_H_ 2.49)/H-9 (δ_H_ 2.22), H-10/H-9, and H-5 (δ_H_ 7.01)/H-10/H-9 demonstrated that H-5, H-10, H-9, and Me-15 were cofacial and β-oriented. On the contrary, the α-orientations of H-4 (δ_H_ 1.84) and H-11 (δ_H_ 2.12) were detected by the NOESY correlations of Me-13 (δ_H_ 0.99)/H-11 and H-11/H-4. Therefore, the structure of **5** with its relative configuration was tentatively established. The absolute configuration was assigned as 1R,4R,10R,9S,11R based on the comparison of calculated and experimental ECD spectra and named Pestalotis E.

Antitumor effects of compounds **1**–**5** were initially tested using a cell counting kit (CCK) to elucidate the inhibitory potency, selectivity, and potential interfering features in vitro. First, the inhibition effect of the compounds on SW 480, LoVo, HuH-7, and MCF-7 was determined through concentration–inhibition curves in vitro with the resulting IC_50_ values shown in Table 3. The result shows that the proliferation of HuH-7, LoVo, and HuH-7 was inhibited by compounds **1** and **4**, among which compound **4** showed the strongest antitumor effect (IC_50_ < 10 μM). Then, the efficacy of compound **4** was tested on more tumor strains, and it was shown that compound **4** had the strongest effect on HuH-7 cell lines.

To identify whether compound **4** could induce HuH-7 cell apoptosis, flow cytometry analyses by Annexin-V-PE and PI double staining assay were performed (Figure 4). After being treated with **4,** compared to the untreated cells, apoptosis rates of HuH-7 were 16.75% with 2.0 μΜ, and 19.38% with 5.0 μM of **4**. These results provide research support for the further cytotoxic study of compound **4**.

## 3. Materials and Methods

### 3.1. General Experimental Procedures

Optical rotations were obtained on a PerkinElmer 341 digital polarimeter. IR spectra were recorded on Shimadzu FTIR-8400S spectrometers. NMR spectra were obtained with a Bruker AV III 600 NMR spectrometer (chemical shift values are presented as δ values with TMS as the internal standard). HR-ESIMS spectra were performed on an LTQ-Orbitrap XL spectrometer. Preparative HPLC was performed on a Lumtech K-1001 analytic LC equipped with two pumps of K-501, a UV detector of K-2600, and an YMC Pack C_18_ column (250 mm × 10 mm, i.d., 5 μM, YMC Co. Ltd., Kyoto, Japan) eluted with CH_3_OH-H_2_O at a flow rate of 2 mL/min. C_18_ reversed-phase silica gel (40~63 μM, Merk, Darmstadt, Germany), MCI gel (CHP 20P, 75~150 μM, Mitsubishi Chemical Corporation, Tokyo, Japan), and silica gel (100~200 mesh, Qingdao Marine Chemical plant, Qingdao, China) were used for column chromatography. Pre-coated silica gel GF254 plates (Zhi Fu Huang Wu Pilot Plant of Silica Gel Development, Yantai, China) were used for TLC. All solvents used were of analytical grade (Beijing Chemical Works).

### 3.2. Fungal Material

The strain used in this work was isolated from the root of the traditional Chinese medicinal plant Ligusticum chuanxiong Hort., which was collected from Sichuan Province, China (2019), and was authenticated by Professor Gongxi Chen (Jishou University). The region ITS1-5.8S-ITS2-28S of the genomic DNA was amplified by PCR and DNA sequencing. Following comparisons with BLAST, we found that this sequence exhibited a significantly high identification with Pestalotiopsis sp. YM312942 18S ribosomal RNA gene, partial sequence; internal transcribed spacer 1, 5.8S ribosomal RNA gene, and internal transcribed spacer 2, complete sequence; and 28S ribosomal RNA gene, partial sequence (GenBank accession number JN868118.1). Based on the comparisons, the fungus was identified as Pestalotiopsis sp. The voucher specimen (CS191126) was deposited at the Institute of Medicinal Plant Development, Chinese Academy of Medical Sciences.

### 3.3. Extraction and Isolation

The strain was cultured on potato dextrose agar (PDA) for 5 days to prepare the seed culture. Agar plugs were inoculated into Erlenmeyer flasks (500 mL), each containing 100 g of rice and 100 mL of water, and each flask had been previously sterilized by autoclaving. All flasks were incubated at 25 °C for five weeks. The fermented rice substrate was extracted three times with EtOAc at room temperature, and the solvent was evaporated under vacuum. EtOAc extracts (27.3 g) were subjected to column chromatography (CC) with silica gel (200–300 mesh) using two gradients (ether-EtOAc, 100:0, 60:1, 20:1, 5:1, and 1:1, v/v) to afford five fractions A-E. Fraction B (783 mg) was chromatographed by semi-preparative HPLC using MeOH-H_2_O (85:15, *v*/*v*) to yield compounds **1** (3.4 mg, t_R_ = 15.6 min) and **2** (4.1 mg, t_R_ = 17.9 min) and **3** (3.1 mg, t_R_ = 25.3 min). Fraction D (5.8 g) was loaded on an ODS C_18_ column (2 × 80 cm) eluted with MeOH-H_2_O (60:40; 70:30; 80:20; 100:0, v/v) to give five subfractions (Fr. D1-D4). Subfraction D3 (462 mg) was chromatographed by semi-preparative HPLC using MeOH-H_2_O (80:20, v/v) to yield compound **5** (6.1 mg, t_R_ = 19.5 min). Fraction E (513 mg) was purified through preparative HPLC elution using an MeOH-H_2_O (65: 35, v/v) system to give compound **4** (7.9 mg, t_R_ = 31.7 min).

*Pestalotis* A (**1**): C_15_H_20_O_5_, white amorphous powder; [α${]}_{D}^{20}$+ 19.0 (c = 0.1, MeOH); IR (KBr) νmax: 3401, 1646, 1633 cm^−1^; CD (MeOH, Δε) λmax 219 (-8.89), 235 (+1.30); HR-ESI-MS m/z 303.1210 [M + Na]^+^ (calcd 303.1203); ^1^H and ^13^C-NMR spectra data, see Table 1.

*Pestalotis* B (**2**): C_15_H_22_O_5_, white amorphous powder; [α${]}_{D}^{20}$+ 23.2 (c = 0.1, MeOH); IR (KBr) νmax: 3446, 1743, 1739 cm^−1^; CD (MeOH, Δε) λmax 218 (-20.74), 248 (+0.79); HR-ESI-MS m/z 305.1387 [M + Na]^+^ (calcd 305.1359); ^1^H and ^13^C-NMR spectra data, see Table 1.

*Pestalotis* C (**3**): C_15_H_26_O_5_, white amorphous powder; [α${]}_{D}^{20}$+ 17.2 (c = 0.1, MeOH); IR (KBr) νmax: 3484, 1653, 1651 cm^−1^; CD (MeOH, Δε) λmax 232 (-3.92), 257 (+0.93); HR-ESI-MS m/z 307.1500 [M + Na]^+^ (calcd 307.1516); ^1^H and ^13^C-NMR spectra data, see Table 1.

*Pestalotis* D (**4**): C_17_H_25_NO_5_, white amorphous powder; [α${]}_{D}^{20}$+ 24.5 (c = 0.1, MeOH); IR (KBr) νmax: 3470, 3446, 1743 cm^−1^; CD (MeOH, Δε) λmax 239 (+3.95), 263 (+5.35), 289 (+12.13); HR-ESI-MS m/z 346.1627 [M + Na]^+^ (calcd 346.1625); ^1^H and ^13^C-NMR spectra data, see Table 2.

*Pestalotis* E (**5**): C_17_H_26_O_4_, white amorphous powder; [α${]}_{D}^{20}$+ 13.1 (c = 0.1, MeOH); IR (KBr) νmax: 3462, 3421, 1710 cm^−1^; CD (MeOH, Δε) λmax 208 (+5.99), 220 (+5.94), 238 (+7.51), 270 (-0.09), 337 (+0.81); HR-ESI-MS m/z 317.1725 [M + Na]^+^ (calcd 317.1723); ^1^H and ^13^C-NMR spectra data, see Table 2.

### 3.4. Experimental Procedures for Bioassay

#### 3.4.1. Cell Culture

The human colon cancer SW-480 cell line, obtained from Cell Resource Center, Institute of Basic Medical Science, CAMS (Beijing, China), was routinely maintained in MEM supplemented with 10% FBS and 1% NEAA at 37 °C in a humidified atmosphere containing 5% CO_2_.

The human colon cancer LoVo cell line, obtained from Cell Resource Center, Institute of Basic Medical Science, CAMS (Beijing, China), was routinely maintained in F12-K supplemented with 10% FBS 37 °C in a humidified atmosphere containing 5% CO_2_.

The human liver cancer HuH-7 cell line, obtained from Cell Resource Center, Institute of Basic Medical Science, CAMS (Beijing, China), was routinely maintained in MEM supplemented with 10% FBS and 1% NEAA at 37 °C in a humidified atmosphere containing 5% CO_2_.

The human breast cancer MCF-7 cell line, obtained from Cell Resource Center, Institute of Basic Medical Science, CAMS (Beijing, China), was routinely maintained in MEM supplemented with 10% FBS 37 °C in a humidified atmosphere containing 5% CO_2_.

#### 3.4.2. Cell Viability Assay

The cell viability was evaluated using a cell counting kit (CCK-8, Dojindo Molecular Technologies Inc., Japan). Briefly, cells were seeded into 96-well plates (Costar, US) at a density of 1 × 10^4^ cells/well (n = 3). Cells were incubated at 37 °C for 24 h, and then the supernatant was removed. Subsequently, cells were exposed to different extractions at various concentrations, followed by further incubation at 37 °C. The supernatant was then removed again, followed by incubation with 90 µL complete medium and 10 µL CCK-8 solution at 37 °C. Proliferation activity was calculated by measuring optical density (OD) at a wavelength of 450 nm using a microplate reader (TECAN, Männedorf, Switzerland). 

#### 3.4.3. Statistical Analysis

All data were expressed as means ± standard deviation (SD). One-way ANOVA followed by LSD test was performed for cell experimental comparisons. Most of the data were analyzed using the IBM SPSS Statistics 25.0 software. A *p* < 0.05 was considered statistically significant.

#### 3.4.4. Cell Cycle Assays

Cell cycle analysis was carried out by flow cytometry. First, 2–5 × 10^6^ HuH-7 cells were collected after treatment with 0, 2.0, or 5.0 μM compound **4** for 48 h, and then fixed in 70% precooled ethanol for 90 min after washing thrice with PBS. Subsequently, cells were incubated with RNase A and propidium iodide (PI) at 37 °C for 30 min. Lastly, cells were passed through 70 μm Falcon Filters for single-cell suspension, and cell cycle was analyzed by utilizing flow cytometry.

## 4. Conclusions

The chemical investigation of the EtOAc extract of an endophytic fungus Pestalotiopsis sp. led to the isolation of five new terpenes, named Pestalotis A-E (**1–5**). The structures were established by extensive analyses of spectroscopic data (1D and 2D NMR, HRESIMS) and ECD spectra (Appendix A). Their absolute configurations were established through the comparison of experimental and calculated ECD spectra. Compound **4** exhibited remarkable antitumor activities against HuH-7 with IC_50_ values of 9.35 µM, which were comparable to those of the positive control 5-FU. Moreover, assays demonstrated that compound **4** could induce HuH-7 cell apoptosis.

## Figures and Tables

**Figure 1 molecules-26-07229-f001:**
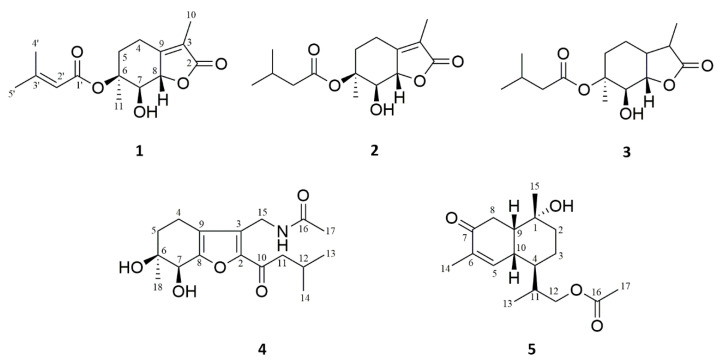
Structures of compounds **1**–**5**.

**Figure 2 molecules-26-07229-f002:**
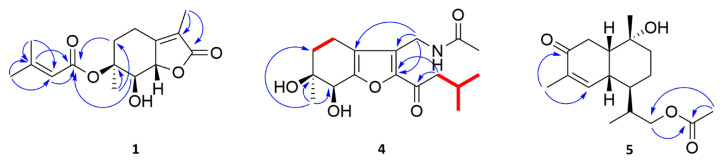
^1^H-^1^H COSY (

) and HMBC (
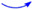
) correlations of compounds **1**, **4** and **5**.

**Figure 3 molecules-26-07229-f003:**
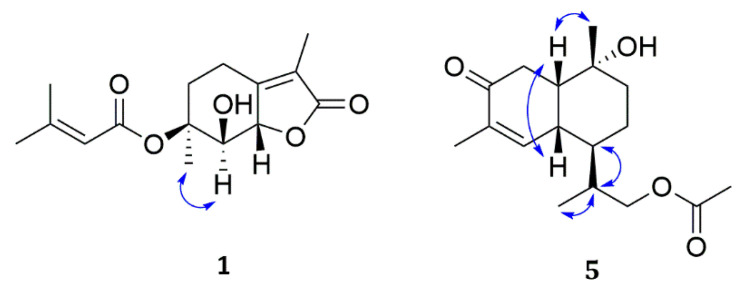
Key NOESY correlations of compounds **1** and **5**.

**Figure 4 molecules-26-07229-f004:**
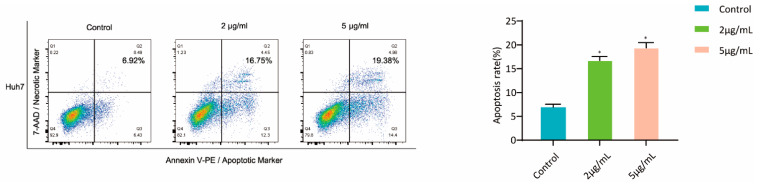
Annexin V PE/7-ADD stained apoptotic cells induced by compound **4** in HuH-7 cells. All data shown represent the means ± S.D. of 3 independent experiments. ^∗^
*p* < 0.05.

**Table 1 molecules-26-07229-t001:** ^1^H (600 MHz) and ^13^C-NMR (150 MHz) assignments of **1**–**3** (DMSO-*d*_6_).

No.	1		2		3	
	*δ* _C_	*δ*_H_ (*J* in Hz)	*δ* _C_	*δ*_H_ (*J* in Hz)	*δ* _C_	*δ*_H_ (*J* in Hz)
2	174.0		174.0		177.3	
3	120.1		120.1		40.9	2.11, m
4	20.9	2.63, m	20.9	2.63, m	23.7	1.38, m
		2.45, m				
5	36.9	1.87, m	36.8	2.26, m	33.9	1.60, m
		1.54, m				1.33, m
6	71.0		71.8		69.5	
7	78.8	4.51, d, 9.6	79.5	4.50, d, 9.6	71.6	3.26, d, 3.6
8	80.7	4.97, d, 9.6	80.7	4.96, d, 9.6	71.9	4.96, d, 3.6
9	160.0		160.0		36.9	1.90, m
10	8.3	1.74, s	8.3	1.73, s	15.3	0.97, d, 6.6
11	25.2	1.03, s	24.3	1.05, s	25.1	1.10, s
1′	165.0		171.7		171.5	
2′	115.1	5.81, s	42.7	2.33, m	42.9	2.25, m
3′	158.5		25.3	2.08, m	24.8	1.27, m
4′	20.1	2.14, s	22.2	0.95, d, 6.6	22.1	0.91, m
5′	27.0	1.93, s	22.1	0.94, d, 6.6	22.1	0.91, m

**Table 2 molecules-26-07229-t002:** ^1^H (600 MHz) and ^13^C-NMR (150 MHz) assignments of **4** (DMSO-*d*_6_) and **5** (CD_3_OD).

No.	4		5	
	*δ* _C_	*δ*_H_ (*J* in Hz)	*δ* _C_	*δ*_H_ (*J* in Hz)
1			71.8	
2	147.6		34.7	1.69, m
				1.45, m
3	126.4		23.1	1.54, m
				1.34, m
4	17.6	2.51, m	40.1	1.84, m
		2.30, m		
5	32.8	1.84, m	151.2	7.01, d, 6.0
		1.59, m		
6	70.4		136.7	
7	68.6	4.22, s	202.6	
8	154.8		36.4	2.63, m
9	121.6		47.1	2.22, m
10	190.4		38.6	2.49, m
11	47.1	2.70, dd, 1.8, 7.2	33.7	2.12, m
12	24.1	2.03, m	69.4	2.41, m
13	22.5	0.92, d, 2.4	11.1	0.99, d, 7.2
14	22.4	0.91, d, 2.4	16.2	1.77, s
15	56.9	5.28, d, 13.2	27.6	1.30, s
		5.22, d, 13.2		
16	170.1		173.1	
17	20.4	2.03, s	21.0	2.01, s
18	24.9	1.13, s		

**Table 3 molecules-26-07229-t003:** In vitro antitumor activity of compounds.

Compounds	IC_50_ (μM)			
	SW480	LoVo	HuH-7	McF-7
1	14.3 ± 2.1 ^a^	13.8 ± 1.3	19.1 ± 4.8	31.5 ± 3.4
2	23.4 ± 2.0	37.0 ± 3.2	43.6 ± 1.2	37.2 ± 1.5
3	25.6 ± 2.1	37.2 ± 1.6	33.3 ± 1.8	>100
4	15.0 ± 1.7	17.2 ± 1.8	9.3 ± 2.0	15.5 ± 1.4
5	32.3 ± 2.8	28.3 ± 2.0	20.1 ± 1.6	>100
5-FU ^b^	1.2 ± 0.1	1.1 ± 0.1	1.3 ± 0.1	0.8 ± 0.1

^a^ The values presented are the means ± SD of triplicate experiments. ^b^ Positive control substance.

## Data Availability

All data and figures in this study are openly available.

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
