# Peer review of "Five New Terpenes with Cytotoxic Activity from Pestalotiopsis sp."

_molecules, 2021, doi:10.3390/molecules26237229_

Round 1

Reviewer 1 Report

The authors report on five new terpenes with cytotoxic activity from Pestalotiopsis sp.

This is an interesting report of some novel terpenoids from this fungus. The bioactivity of one of these compounds seems to be interesting but only a limited number of experiments have been done to characterize these activities. The description of the experiments need to be more precise.

I have a couple of concerns which need to be addressed by the authors.

The configuration of compound 1 is not clear to me and I would argue against it. Both NOESY and the large coupling constant of H-7 would better fit to an alpha-position of H-7. Please comment and give more convincing arguments in the text.

In general: How many times have the experiments been repeated? How often have the apoptosis experiments been reproduced? What are the standard deviations of the results of the flow cytometry?

Table 3: please be realistic. The two digits after the decimal point are not reproducible, therefore, the numbers with only one decimal should be enough here, giving the degree of standard deviations.

How was the fungal strain identified? The word of Professor Gongxi Chen is here not good enough.

Author Response

Dear Editor and Reviewers,

Thank you for your comments concerning our manuscript entitled “Five new terpenes with cytotoxic activity from Pestalotiopsis sp.” (Manuscript ID: 1446842). These comments are valuable and helpful for revising and improving our paper, as well as the important guiding significance to our researchers. We have studied comments carefully and made correction accordingly. Revised portion are marked red in the paper. The detailed corrections in the paper and the responds to the reviewer’s comments are listed as flowing.

Sincerely yours,

Prof. Guoxu Ma

Institute of Medicinal Plant Development, Peking Union Medical College and Chinese Academy of Medical Sciences, Beijing 100193, China;

Tel./fax: 86-010-5783-3296.   

E-mail address: [email protected]

Responds to the reviewer’s comments:

1.

The configuration of compound 1 is not clear to me and I would argue against it. Both NOESY and the large coupling constant of H-7 would better fit to an alpha-position of H-7. Please comment and give more convincing arguments in the text.

Response: Thank you for your advice. The large coupling constant of H-7/H-8 (J = 9.6 Hz) by the NOESY correlations indicated an α-configuration for H-7. We added the NOESY descriptions of H-7 in the manuscript. (We renumbered the carbon atom of compounds 1-5 , and changed H-7a into H-8.)

  1.  

How many times have the experiments been repeated? How often have the apoptosis experiments been reproduced? What are the standard deviations of the results of the flow cytometry?

Response: Thanks for your suggestions. We conducted 3 independent experiments in a row, which compared to control *P<0.05, ∗∗P<0.01, and we added remarks below figure 4.

  1.  

Table 3: please be realistic. The two digits after the decimal point are not reproducible, therefore, the numbers with only one decimal should be enough here, giving the degree of standard deviations.

Response: Thanks. We modified it and kept the value in Table 3 with only one decimal place.

  1.  

How was the fungal strain identified? The word of Professor Gongxi Chen is here not good enough.

Response: Thanks for your suggestions, and we added the content of strain identification in the manuscript.

Reviewer 2 Report

The authors describe the identifiaction of 5 novel compounds in Pestalotiopsis. 3 monoterpenes, 1 sesquiterpene and 1 other compound.

The report of the structures is well supported, however the introduction and conculsion are rather short, are there similar compounds described in fungi of the same genus or a higher phylogenetic order?

Line 180: What University and Institute is Prof. Gongxi Chen associated with?

Table 1: Can you renumber the carbon atoms to avoid nomenclature such as 7 and 7a?

Line 235/247: What solvent was used for the compounds and was a solvent control done?

Author Response

Dear Editor and Reviewers,

Thank you for your comments concerning our manuscript entitled “Five new terpenes with cytotoxic activity from Pestalotiopsis sp.” (Manuscript ID: 1446842). These comments are valuable and helpful for revising and improving our paper, as well as the important guiding significance to our researchers. We have studied comments carefully and made correction accordingly. Revised portion are marked red in the paper. The detailed corrections in the paper and the responds to the reviewer’s comments are listed as flowing.

Sincerely yours,

Prof. Guoxu Ma

Institute of Medicinal Plant Development, Peking Union Medical College and Chinese Academy of Medical Sciences, Beijing 100193, China;

Tel./fax: 86-010-5783-3296.   

E-mail address: [email protected]

Responds to the reviewer’s comments:

1.

The report of the structures is well supported, however the introduction and conculsion are rather short, are there similar compounds described in fungi of the same genus or a higher phylogenetic order?

Response: Thanks for your suggestions. We added descriptions of similar terpenoids in the Pestalotiopsis sp.

  1.  

Line 180: What University and Institute is Prof. Gongxi Chen associated with?

Response: Thanks. We added the information of Prof. Gongxi Chen’s research institutions in the manuscript.

  1.  

Table 1: Can you renumber the carbon atoms to avoid nomenclature such as 7 and 7a?

Response: Thanks for your suggestions, and we renumbered the carbon atoms of compounds 1-5 in the whole manuscript and tables.

  1.  

Line 235/247: What solvent was used for the compounds and was a solvent control done?

Response: Thank you for your advice. In cytotoxic activity experiments, dimethyl sulfoxide (DMSO) was used to dissolve the compounds and the blank control group.

Round 2

Reviewer 1 Report

In their rebuttal the authors answered to all suggestions but these changes were not always incorporated into the text of the manuscript.

The configuration of 3 at C-8 is not clear from the drawing. Please draw the ester moiety in beta-position.

Table 3 still has two decimals. Please reduce it to only one.

Please add the s. d. to figure 4.